# High CSF thrombin concentration and activity is associated with an unfavorable outcome in patients with intracerebral hemorrhage

Harald Krenzlin[1]*, Christina Frenz[1], Jan Schmitt[1], Julia Masomi-Bornwasser[1], Dominik Wesp[1], Darius Kalasauskas[1], Thomas Kerz[1], Johannes Lotz[2], Beat Alessandri[1], Florian Ringel[1], Naureen Keric[1]

**1** Department of Neurosurgery, University Medical Center Mainz, Mainz, Germany, **2** Institute of Clinical Chemistry and Laboratory Medicine, University Medical Center Mainz, Mainz, Germany

* harald.krenzlin@unimedizin-mainz.de

## Abstract

### Background

The cerebral thrombin system is activated in the early stage after intracerebral hemorrhage (ICH). Expression of thrombin leads to concentration dependent secondary neuronal damage and detrimental neurological outcome. In this study we aimed to investigate the impact of thrombin concentration and activity in the cerebrospinal fluid (CSF) of patients with ICH on clinical outcome.

### Methods

Patients presenting with space-occupying lobar supratentorial hemorrhage requiring extraventricular drainage (EVD) were included in our study. The CSF levels of thrombin, its precursor prothrombin and the Thrombin-Antithrombin complex (TAT) were measured using enzyme linked immune sorbent assays (ELISA). The oxidative stress marker Superoxide dismutase (SOD) was assessed in CSF. Initial clot size and intraventricular hemorrhage (IVH) volume was calculated based on by computerized tomography (CT) upon admission to our hospital. Demographic data, clinical status at admission and neurological outcome were assessed using the modified Rankin Scale (mRS) at 6-weeks and 6-month after ICH.

### Results

Twenty-two consecutive patients (9 females, 11 males) with supratentorial hemorrhage were included in this study. CSF concentrations of prothrombin ($p < 0.005$), thrombin ($p = 0.005$) and TAT ($p = 0.046$) were statistical significantly different in patients with ICH compared to non-hemorrhagic CSF samples. CSF concentrations of thrombin 24h after ICH correlated with the mRS index after 6 weeks ($r^2 = 0.73$; $< 0.005$) and 6 months ($r^2 = 0.63$; $< 0.005$) after discharge from hospital. Thrombin activity, measured via TAT as surrogate parameter of coagulation, likewise correlated with the mRS at 6 weeks ($r^2 = 0.54$; $< 0.01$) and 6 months ($r^2 = 0.66$; $< 0.04$). High thrombin concentrations coincide with higher SOD levels 24h after ICH ($p = 0.01$).

**Data Availability Statement:** All relevant data are within the paper and its Supporting Information files.

**Funding:** The funders had no role in study design, data collection and analysis, decision to publish, or preparation of the manuscript. The authors received no specific funding for this work.

**Competing interests:** The authors have declared that no competing interests exist.

## Conclusion

In this study we found that initial thrombin concentration and activity in CSF of ICH patients did not correlate with ICH and IVH volume but are associated with a poorer functional neurological outcome. These findings support mounting evidence of the role of thrombin as a contributor to secondary injury formation after ICH.

## Background

Intracerebral hemorrhage (ICH) accounts for 10–15% of all strokes worldwide [1]. With an annual incidence of 10–30 per 100,000 population, it represents a major public health burden [1]. It is associated with considerable mortality and long-term morbidity [2]. Neuronal damage after ICH is caused by primary tissue disruption and secondary progression due to detrimental processes in the perihematomal zone. Despite ongoing clinical and preclinical research efforts, the underlying pathophysiological mechanisms of secondary deterioration are poorly understood. So far, good prognostic factors and specific treatment targets are missing.

Recently, the importance of the blood coagulation factor IIa (thrombin, FIIa) as key effector after traumatic brain injury and hemorrhagic stroke has become evident [3–5]. Thrombin signaling is mostly mediated through the family of protease-activated receptors (PARs) causing inflammatory responses, cell proliferation/modulation, cell protection and apoptosis [6, 7]. The majority of prothrombin is produced in the liver and, due to its large, spherical shape, it is unable to pass the blood-brain barrier (BBB) [8]. Nevertheless, thrombin and associated factors such as Anti-thrombin III (AT III) have also been detected throughout the central nervous system under physiological conditions [9]. Within the central nervous system (CNS) thrombin exerts both protective and detrimental effects. In pico- to nanomolar ranges (10pM– 10nM), thrombin is protective against a variety of cellular insults, such as glucose deprivation, reactive oxygen species (ROS) or edema formation after ICH [10, 11]. In high concentrations (100 nM–10 μM) thrombin increases edema by TNF-α up-regulation, neuronal damage and death after ischemia in mice [12–14]. These processes lead to extended neuronal injury predominantly mediated via protease-activated receptor-1 signaling [15]. Interestingly, the majority of perihematomal thrombin accumulation has been linked to neuronal expression, rather than systemic influx [16]. The CNS is the only known site of extra-hepatic thrombin production [17]. Neuronal cell loss within the tissue at risk and neurological deficits are associated with higher concentrations of thrombin in mice [16]. This supports the potential role of thrombin as a target of novel therapeutic regimens in the treatment of intracerebral hemorrhage. As human perihematomal tissue samples are not readily available, other specimens need to be analyzed to gain insight into the role of thrombin after ICH. It has previously been shown that thrombin and its inactive precursor prothrombin are detectable in human CSF [5]. However, no published data are available addressing thrombin within CSF after ICH.

We hypothesized that thrombin accumulation occurs in all compartments including the cerebrospinal fluid (CSF) after ICH and, in higher concentrations, might contribute to detrimental neurological outcome. In this study, prothrombin, thrombin and TAT were analyzed in the CSF of ICH patients and correlated with their functional neurological outcome.

## Methods

### Patient population

From February 2017 to February 2019, 20 consecutive patients (9 female and 11 male) that required extra-ventricular drainage due to space-occupying supratentorial ICH with

ventricular hemorrhage were included in our study. Age ranged from 40 to 80 years (66±12 years). All patients had one or more underlying conditions. Arterial hypertension was most common and present in 75% of all patients, chronic heart disease (35%) and malignancies (30%) ranked 2nd and 3rd (S1 Table). All patients received standard intensive care medical treatment according to current clinical guidelines. EVD was established on admission on our neurosurgical intensive care unit. Patients were followed up until six months after discharge or until death occurred. CSF and clinical data were collected and analyzed prospectively. All patients received a thorough clinical examination on admission and before discharge from our hospital. CT was used to analyze hematoma volume and localization as described before [18]. The clinical data and baseline characteristics are summarized in Table 1. Each patient's clinical status was graded according to the modified Rankin scale (mRS) and Glasgow coma scale (GCS) prior to admission. The neurological outcome was measured using the mRS and Glasgow outcome scale extended (GOSE) at 6 weeks and 6 months after ICH occurrence. CSF from patients with normal pressure hydrocephalus was collected and served as controls.

## Sampling procedure

CSF was obtained through extra-ventricular drainage at day 1 and 3 after ICH onset. Concordant blood samples were obtained via an arterial canula. CSF of controls were obtained from either lumbar drainages placed for normal pressure hydrocephalus evaluation or from intraoperative opening of CSF spaces in meningioma and schwannoma patients. Samples were collected in a sterile plastic tube. The tube was centrifuged at 1500 G at 4°C for 5 min. The supernatant was frozen, and aliquots were stored at -80°C until analysis.

## Measurement of analytes in CSF and blood samples

The prothrombin (NBP2-60624, Novus Biologicals, Abingdon, UK), thrombin (NBP2-60590, Novus Biologicals, Abingdon, UK) and TAT (NBP2-60605, Novus Biologicals, Abingdon, UK)

**Table 1. Baseline demographics and patients characteristics.**

|  | ICH patients | Control subject |
|---|---|---|
| No of subjects | 22 | 4 |
| Mean age (SD) | 65.6 (12.3) | 67.0 (5.7) |
| Sex |  |  |
| Female | 8 | 2 |
| Male | 14 | 2 |
| ICH score | 3 (1) |  |
| Hematoma localization |  |  |
| frontal | 12 | n.a. |
| parietal | 8 | n.a. |
| occipital | 0 | n.a. |
| Basal ganglia | 2 | n.a. |
| Clot volume (cm$^3$) (SD) | 26.61 (29.93) | n.a. |
| Intraventricular clot (cm$^3$) (SD) | 76.96 (94.88) | n.a. |
| Perihematomal zone (cm3) (SD) | 90.54 (44.27) | n.a. |
| CSF concentration (ng/ml) |  |  |
| Prothrombin | 20.18 (1.4) | 16.68 (1.17) |
| Thrombin | 4.12 (1.3) | 0.86 (0.36) |
| TAT | 1.95 (1.7) | 0.57 (0.11) |
| SOD | 2.05 (0.95) | 1.21 (0.05) |

concentration of each sample was measured with an enzyme-linked immunosorbent assay. SOD and fibrinogen were measured via photometry.

## Measurement of hematoma volume

ICH and IVH volumes were calculated by CT, with a slice thickness of 1 mm. On serial slices in one direction, the intraventricular and intracerebral hematomas were segmented separately, while the volume were calculated using the Brainlab software (Brainlab, Munich, Germany) (Tables 1 and 2).

## Statistical analysis

Findings were reported as mean or median ± SD. For statistical analysis, we used the non-parametric Mann-Whitney U-test. Relations among FII, FIIa, TAT, SOD and clinical outcome parameters two-way analysis of variance (ANOVA) with Tukey's multiple comparison post hoc test were performed using GraphPad Prism version 8.4.2 for macOS, GraphPad Software, La Jolla California USA, www.graphpad.com. A value of $P < 0.05$ was accepted as statistically significant.

## Ethical approval

Data acquisition and analysis was performed in an anonymous fashion and was approved by the Ethics Committees of the medical association of Rhineland Palatinate and Lower Saxony, Germany (837.374.16). According to the local laws, no informed consent is necessary for such kind of analysis.

## Results

### Patient outcome and follow-up

Median GCS on admission was 10 ± 5. The mean ICH score on admission was 3 ± 1. Median historical mRS was 1 ± 1. The mRS deteriorated after ICH to 5 ± 1 at week 6 after ICH. The median mRS remained at 5 ± 2 at 6 months after ICH. (Table 2)

### CSF FII, FIIa and TAT of controls and ICH patients

24h after ICH mean CSF levels of thrombin were 4.12±1.3 ng/ml compared to 0.86±0.36 ng/ml in healthy controls. Likewise, mean levels of prothrombin within the CSF was 20.18±1.4 ng/ml in patients with ICH and 16.68±1.17 ng/ml in controls. Mean TAT levels were 1.95±1.7 ng/ml or 0.57±0.11 ng/ml respectively. A statistically significant difference was evident between CSF concentrations of prothrombin (p = 0.001), thrombin (p = 0.005) and TAT (p = 0.046) in patients with ICH compared to healthy controls. (Table 1) Plasma levels of FII and FIIa did not correlate with their corresponding CSF levels. (Table 3)

**Table 2. Disability and dependence over time (median; IQR).**

|       | Admission / prior to ICH | 6 weeks | 6 month |
|-------|--------------------------|---------|---------|
| GCS   | 10 (3, 14)               | n.a.    | n.a.    |
| mRS   | 1 (0, 1)                 | 5 (3, 5)| 5 (3, 6)|
| GOSE  | n.a.                     | 3 (2, 3)| 1 (1, 3)|

**Table 3.  CSF prothrombin, thrombin and TAT of patients with ICH and controls.**

| | Prothrombin (FII) | | Thrombin (FIIa) | | TAT | |
|---|---|---|---|---|---|---|
| | Spearman's ρ | p | Spearman's ρ | p | Spearman's ρ | p |
| ICH score | 0.69 | 0.005 | 0.68 | 0.004 | 0.04 | 0.45 |
| mRS 6 weeks | 0.3 | 0.28 | 0.61 | 0.004 | 0.65 | 0.013 |
| mRS 6 month | 0.33 | 0.22 | 0.57 | 0.009 | 0.65 | 0.015 |
| Clot size | 0.47 | 0.07 | 0.23 | .35 | 0.01 | 0.53 |
| Plasma / CSF | 0.007 | 0.77 | 0.08 | 0.3 | - | - |

### CSF fibrinogen and SOD after ICH

CSF levels of fibrinogen were 1.57±0.74 ng/ml 24h- and 1.5±0.89 ng/ml 72h after ICH. SOD levels within the CSF were 2.05±0.9 U/l 24h after ICH and 1.21±0.05 U/l in controls.

While initial levels of CSF fibrinogen and thrombin showed no correlation, both were inversely correlated 72h after onset of ICH (r = -0.68; p = 0.01). Higher levels of CSF thrombin correlate with higher levels of SOD 24h after ICH occurrence (r = 0.64; p = 0.095). (Fig 1)

### Correlation of CSF FII, FIIa and TAT with clinical outcome

CSF concentrations of FIIa 24h after ICH correlated with the initial ICH score (r = 0.68; p = 0.004), as well as the mRS disability index 6 weeks (r = 0.61; p = 0.004) and 6 months (r = 0.57; p = 0.009) after discharge from hospital. Further, the TAT-complex as surrogate parameter of coagulation correlated with the mRS disability index at 6 weeks (r = 0.65; p = 0.013) and 6 months (r = 0.65; p = 0.015). (Table 3, Figs 1 and 2) In contrast, Prothrombin, Thrombin and TAT had no statistically significant correlation neither to the initial ICH volume nor to the IVH volume. Likewise, plasma levels of FII, FIIa and TAT did not correlate with the clinical outcome. (Table 3)

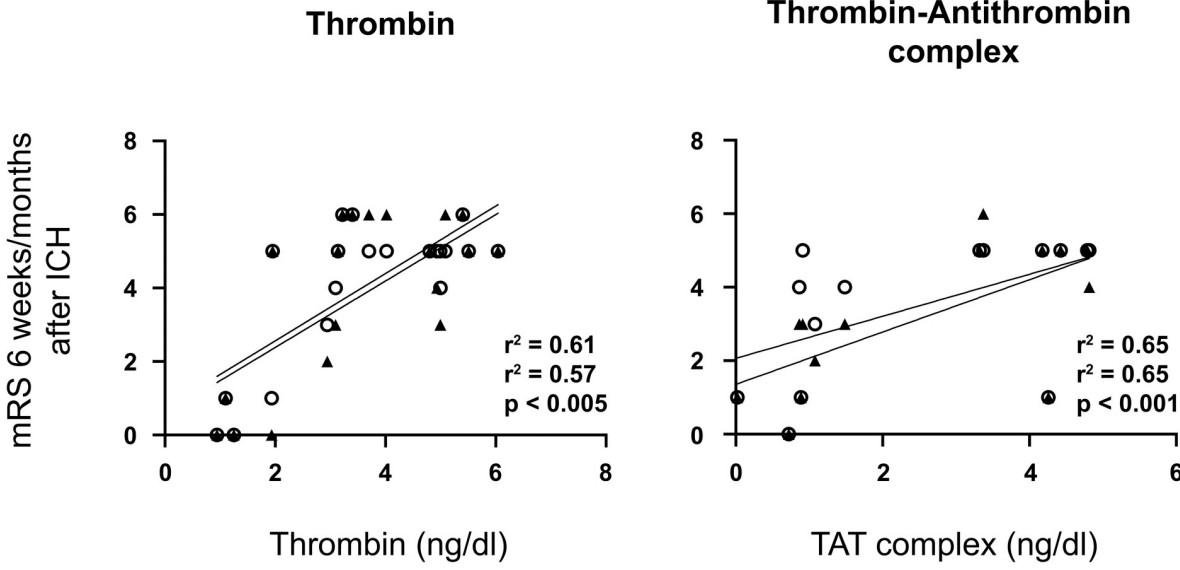

**Fig 1.  mRS after ICH is dependent on CSF FIIa and TAT concentrations.** CSF concentrations of FIIa 24h after ICH correlated with the mRS disability index 6 weeks (r = 0.61; p = 0.004) and 6 months (r = 0.57; p = 0.009) after discharge from hospital. Further, the TAT-complex as surrogate parameter of coagulation correlated with the mRS disability index at 6 weeks (r = 0.65; p = 0.013) and 6 months (r = 0.65; p = 0.015).

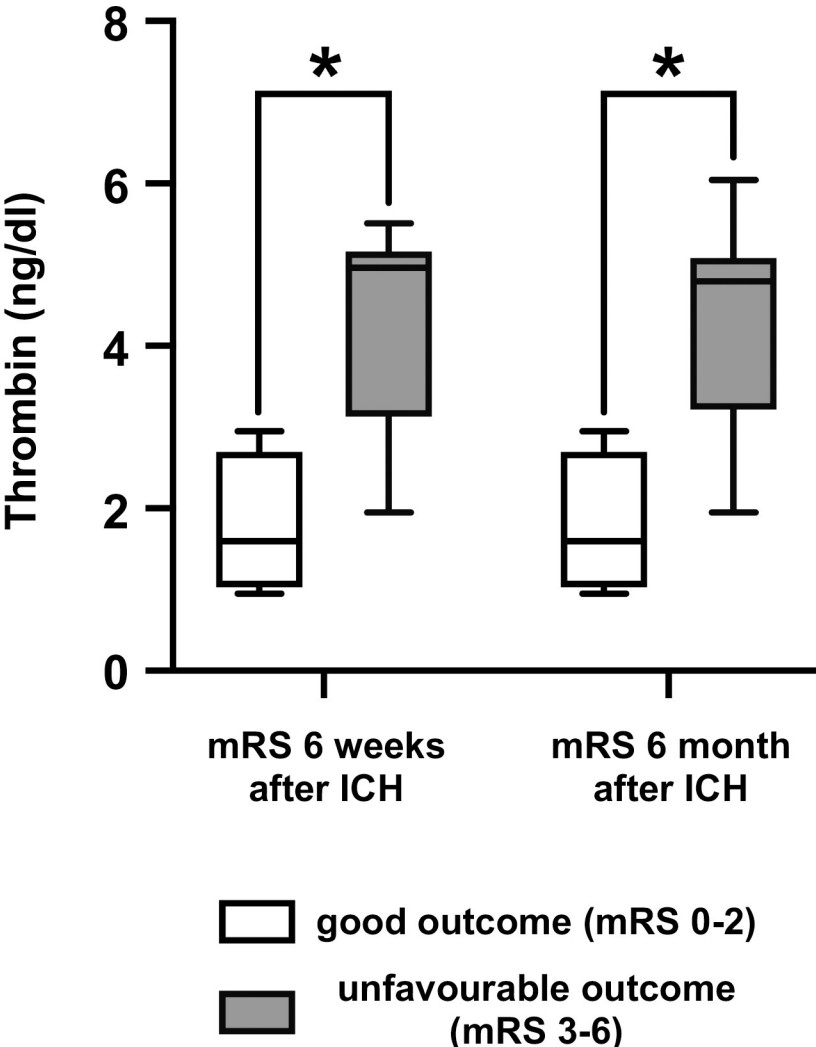

**Fig 2. Higher CSF concentrations of thrombin are associated with an unfavourble outcome.** Higher FIIa CSF concentrations 24h after ICH correlated with an unfavorable outcome after 6 weeks and 6 months.

Correlation of CSF FII, FIIa and TAT with clinical outcome was independent from the type of surgical intervention. In our data, clot evacuation or lysis did not lead to better neurological outcome after 6 weeks or 6 months compared to conservative medical treatment (p = 0.71). High concentrations of FII, FIIa or TAT in CSF did not increase the likelihood of necessary surgical interventions during the acute phase of hospital care.

## Discussion

In our study, higher concentrations of thrombin in the CSF of patients suffering from ICH correlated with an unfavorable outcome at 6 weeks and 6 months after discharge from hospital. We further show that prothrombin and thrombin within the CSF does correlate with the ICH score at the time of admission. This is plausible as thrombin has been shown to be neurotoxic both *in vitro* and *in vivo* if exceeding a threshold of 100nM [19]. Thrombin is inhibited by antithrombin III, resulting in an inactive proteinase/inhibitor complex (TAT). Interestingly, exceeding concentrations of antithrombin III are also detrimental to neuronal health

and might add to adverse thrombin effects [19]. In contrast to thrombin, ATIII mRNA expression has not been detected within normal CNS tissue and is supposed to be largely derived from passage across the blood-brain barrier [20]. It is about 100x higher in plasma, than it is in CSF. In times of BBB disruption, ATIII might enter then CNS and contribute to neuronal damage in the wake of ICH [20]. In our cohort, higher concentrations of thrombin and TAT complex within CSF correlated with a higher mRS at 6 weeks and 6 months after ICH, similar to those of unbound thrombin. These results fall in line with data published from patients with SAH where thrombin activity correlated with the degree of SAH (according to Fisher's CT classification), the persistence of a subarachnoid clot and the development of vasospasm [14]. In SAH, the Thrombin-Anti-Thrombin complex as surrogate of thrombin activity was found to correlate with the Hunt and Hess grading on admission. Although not statistically significant, larger amounts of TAT persisting over a longer period of time were found to be present in patients with worse neurological outcome [15]. In a preclinical study, nude mice developed hydrocephalus after injection of human hemorrhagic CSF [21]. The authors suggest that various acellular components of CSF inducing secondary brain injury and post-hemorrhagic hydrocephalus. However, in our small patient cohort, the severity and volume of ICH and IVH did not correlate neither with thrombin nor with the occurrence of a post-hemorrhagic hydrocephalus proving the complexity of the underlying pathophysiology.

It is speculated that thrombin induced neurotoxicity is mediated via Par-1, one of 4 members of the protease activated receptors family as activation of Par-1 per se has been linked to neuronal cell death [22]. The increases of oxidative stress and reduction of mitochondrial membrane potential might serve as possible explanation how Par-1 activation subsequently leads to cellular toxicity [23]. Another pathway leading to reactive oxygen species involves activation of microglial NADPH oxidase by thrombin induced upregulation of NADPH oxidation proteins gp91, p47-phox and p67-phox [22]. In our study SOD, higher concentration of thrombin coincided with higher concentrations of SOD at day 1, indirectly indicating the increased occurrence of ROS. Further, higher concentrations of CSF thrombin lead to decreased amounts of fibrinogen 3 days after ICH. As Inhibition of fibrin formation reduces neuroinflammation and improves long-term outcome after intracerebral hemorrhage, this might hint at another mechanism leading to secondary damage induction mediated by thrombin [24].

In contrast to our expectations, thrombin did not correlate with ICH or the IVH volume in our study. It remains a matter of debate, whether thrombin is washed in as a result of blood extravasation and BBB breakdown or mainly produced within the CNS as response to various stressors. Local thrombin expression in the perihematomal zone after ICH might contribute to thrombin levels in CSF [16]. As numerous MRI studies suggests the existence of a perihematomal penumbra with functionally impaired, but potentially reversible neuronal injury, this thin rim of 2 mm to a maximum width of 1 cm surrounding the site of injury, might as well be the area most devastatingly influenced by thrombin [16, 25].

Limitations of the study are due to a rather small and heterogeneous group of patients suffering from severe intracerebral hemorrhage. Limited knowledge of events prior to hospital admission and the number of concomitant illnesses might constitute an additional limitation of our study. Further research combining larger patient cohorts and multivariate analysis could lead to more definitive conclusions on the role of the cerebral thrombin system in the development of secondary brain injury after ICH.

## Conclusion

In summary, in the presented data thrombin concentration and activity correlate with the neurological outcome after ICH. Further, generation of ROS seems to be involved in these

processes. Our data adds to a mounting body of evidence hinting at the importance of thrombin as a contributor to secondary injury formation after ICH.

## Supporting information

**S1 Fig. Clot volume is independent from CSF FIIa and TAT concentrations.** CSF concentrations of FIIa (r = 0.13; p = 0.68) and TAT (r = -0.18; p = 0.51) show no correlation with clot volume 24h after ICH. Likewise, no correlation between FIIa (r = 0.36; p = 0.137) and TAT (r = 0.23; p = 0.422) with intraventricular clot volume were found 24h after ICH.
(TIF)

**S1 Table. Underlying conditions.**
(DOCX)

## Author Contributions

**Conceptualization:** Harald Krenzlin, Julia Masomi-Bornwasser, Dominik Wesp, Thomas Kerz, Beat Alessandri, Florian Ringel, Naureen Keric.

**Data curation:** Harald Krenzlin, Christina Frenz, Jan Schmitt, Julia Masomi-Bornwasser, Dominik Wesp, Thomas Kerz, Johannes Lotz.

**Formal analysis:** Harald Krenzlin, Christina Frenz, Darius Kalasauskas, Beat Alessandri, Naureen Keric.

**Project administration:** Harald Krenzlin, Florian Ringel.

**Resources:** Florian Ringel.

**Supervision:** Harald Krenzlin, Dominik Wesp, Thomas Kerz, Beat Alessandri, Florian Ringel, Naureen Keric.

**Validation:** Naureen Keric.

**Visualization:** Harald Krenzlin.

**Writing – original draft:** Harald Krenzlin.

**Writing – review & editing:** Julia Masomi-Bornwasser, Darius Kalasauskas, Johannes Lotz, Beat Alessandri, Florian Ringel, Naureen Keric.

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
