## [Decision Letter · Decision Letter 0]

10 Sep 2020

PONE-D-20-21960

High CSF thrombin concentration and activity is associated with an unfavorable outcome in patients with intracerebral hemorrhage

PLOS ONE

Dear Dr. Krenzlin,

Thank you for submitting your manuscript to PLOS ONE. After careful consideration, we feel that it has merit but does not fully meet PLOS ONE’s publication criteria as it currently stands. Therefore, we invite you to submit a revised version of the manuscript that addresses the points raised during the review process.

We look forward to receiving your revised manuscript.

Kind regards,

Tamil Selvan Anthonymuthu, Ph. D

Academic Editor

PLOS ONE

Journal Requirements:

3.Thank you for stating the following financial disclosure:

 [The funders had no role in study design, data collection and analysis, decision to publish, or preparation of the manuscript.].

4. Please ensure that you refer to Figure 2 in your text as, if accepted, production will need this reference to link the reader to the figure.

Additional Editor Comments (if provided):

It would be better to indicate the exact p value rather than showing the range such as p< 0.01, p<0.05, p<0.09 in Table 3 and in the subsequent text.

Reviewers' comments:

Reviewer's Responses to Questions

**Comments to the Author**

1. Is the manuscript technically sound, and do the data support the conclusions?

Reviewer #1: Yes

Reviewer #2: Yes

2. Has the statistical analysis been performed appropriately and rigorously? 

Reviewer #1: I Don't Know

Reviewer #2: Yes

3. Have the authors made all data underlying the findings in their manuscript fully available?

Reviewer #1: Yes

Reviewer #2: Yes

4. Is the manuscript presented in an intelligible fashion and written in standard English?

Reviewer #1: Yes

Reviewer #2: Yes

5. Review Comments to the Author

Reviewer #1: The authors report high CSF thrombin concentration/activity is associated with an unfavorable outcome with intracerebral hemorrhage, and report thrombin concentration/activity in the CSF of patients with ICH on clinical outcomes in patients presenting with space-occupying lobar supratentorial hemorrhage requiring extra-ventricular drainage (EVD). They measured CSF levels of thrombin, prothrombin, thrombin-antithrombin complexes (TATs), and superoxide dismutase (SOD) in CSF. Initial clot size and intraventricular blood volume were calculated by computerized tomography (CT) on admission, and demographic data, clinical status at admission and neurological outcome were assessed using the modified Rankin Scale (mRS) at 6-weeks and 6-months in 22 consecutive patients (9 females, 11 males) with supratentorial hemorrhage. CSF concentrations of prothrombin (p < 0.005), thrombin (p = 0.005) and TAT (p = 0.046) were statistically different in patients with ICH compared to non-hemorrhagic CSF samples. CSF concentrations of thrombin 24h after ICH correlated with the mRS index after 6 weeks (r2 = 0.73; < 0.005) and 6 months (r2 = 0.63; < 0.005) after discharge from hospital. Thrombin activity, measured via TAT as surrogate parameter of coagulation, likewise correlated with the mRS at 6 weeks (r2 = 0.54; < 0.01) and 6 months (r2 = 0.66; < 0.04). High thrombin concentrations coincide with higher SOD levels 24h after ICH (p = 0.01). They conclude that increased thrombin levels and thrombin activity are associated with a poorer neurological outcome after ICH, and have the potential to serve as predictive markers for individual treatment decision making and future novel therapeutic targets.

Overall, the data and presentation are interesting. The authors are to be congratulated for their interesting study.

Comments:

1. From this reviewer’s perspective, it follows that the larger the clot burden and size of the intracranial hemorrhage, the more probability is that levels of thrombin and other thrombin generation thrombin activity biomarkers will be higher. However, as you thoughtfully noted, intraventricular volumes on CT scans did not correlate with your various biomarkers of thrombin activity in particular thrombin levels. I still think it would be helpful to include a scattergram correlating volume with thrombin levels. Also, has this finding been previously reported?

2. You state CSF fibrinogen and thoughtfully have control comparisons. Is there any correlation between plasma and CSF fibrinogen levels?

3. I’m surprised how small your standard deviation values are for your biomarkers. These are standard deviation and not the standard error of the means, correct?

4. You use the term ictus, I think to describe stroke or CNS event. I think another descriptive term would be better.

Additional comments:

Results

1. You state, “Serum levels of FII and FIIa did not correlate with their corresponding CSF levels”. This should be plasma, not serum. You also state this later on as serum levels. Again, coagulation factor levels are obtained from plasma.

2. You state , “Further, the TAT-complex as surrogate parameter of coagulation and fibrinolysis correlated with the mRS disability index at 6 weeks (2 = 0.65; p < 0.01) and 6 months (r = 0.65; p < 0.01)”. How does thrombin – antithrombin complexes provide any surrogate parameter of fibrinolysis? Please specifically explain if not remove the term fibrinolysis.

Discussion

1. Your first paragraph restates information that really should be part of your introduction. My suggestion is in your first paragraph state the novelty and important information from your interesting study, and include your second paragraph is your first paragraph.

2. In your first paragraph, you state, “While small amounts of activated coagulation FII”- be consistent as you described this activated coagulation factor to is thrombin. Is this what you’re referring to? Please be consistent.

3. In your second paragraph, you state, Thrombin is inhibited by antithrombin III, resulting in an inactive proteinase/inhibitor complex (TAT). Interestingly, exceeding concentrations of antithrombin III are also detrimental to neuronal health and might add to adverse thrombin effects.[28]” Antithrombin is primarily a circulating plasma anticoagulant. How does antithrombin get into the CSF? What are the normal levels of antithrombin in the CSF? My understanding is that antithrombin in antithrombin – thrombin complexes are due to active bleeds and breakdown of the blood-brain barrier. Please add this information to your discussion.

4. You state, “In our cohort, higher concentrations of thrombin and TAT complex within CSF correlated with a higher mRS at 6 weeks and 6 months after ICH, similar to those of unbound thrombin. These results fall in line with data published from patients with SAH where thrombin activity correlated with the degree of SAH (according to Fisher’s CT classification), the persistence of a subarachnoid clot, and the development of vasospasm.[29]” as previously stated, at least providing a box plot correlation of the intracranial hemorrhage with thrombin generation in your study would be useful.

Reviewer #2: The introduction could be shortened by 50%.

Probably should report variables as median with interquartile range (rather than standard deviation). It's a bit more informative.

How were controls selected? Were they matched for any demographics or comorbidities with patients? If not, how might this limit your results? How was CSF obtained from controls? How might different sampling methods affect your results?

I could not find mention of whether any patients had intraventricular hematoma, and if so, how much?

Need consistency in the number of decimal places used to report out p-values in tables.

Please report mean/median time to CSF sample collection from ICH ictus, if you have it, or list this as a limitation.

Given the very small sample size, the conclusion of both the abstract and the manuscript are over-stated. It is possible that CSF thrombin may be associated with outcome, but it's also quite possible that a larger, better powered, more random sample would prove your current findings wrong. Or that a larger study would find that only certain patients (perhaps with certain hematomas) demonstrate the associations that you found. Alot left to learn and your data are very preliminary.

What do you think of your findings in the context of the translational science work using preclinical SAH models - Wan, S., Wei, J., Hua, Y. et al. Cerebrospinal Fluid from Aneurysmal Subarachnoid Hemorrhage Patients Leads to Hydrocephalus in Nude Mice. Neurocrit Care (2020). https://doi.org/10.1007/s12028-020-01031-0?

Overall, nicely done.

6. PLOS authors have the option to publish the peer review history of their article (what does this mean?). If published, this will include your full peer review and any attached files.

Reviewer #1: No

Reviewer #2: No

---

## [Author Response · Author response to Decision Letter 0]

12 Oct 2020

• rebuttal letter that responds to each point raised by the academic editor and reviewer(s) (upload this letter as a separate file labeled 'Response to Reviewers')

• A marked-up copy of your manuscript that highlights changes made to the original version. (upload this as a separate file labeled 'Revised Manuscript with Track Changes')

• An unmarked version of your revised paper without tracked changes (upload this as a separate file labeled 'Manuscript')

2. Journal Requirements:

• ensure that your manuscript meets PLOS ONE's style requirements, including those for file naming. The PLOS ONE style templates can be found at

• Please include your tables as part of your main manuscript and remove the individual files. Please note that supplementary tables (should remain/ be uploaded) as separate "supporting information" files. 

Response: Table 1 has now been included with the main file. Supplementary Table 1 remained as individual file and has been uploaded as supporting information" file.

• Address the following queries: 

Please clarify the sources of funding (financial or material support) for your study. List the grants or organizations that supported your study, including funding received from your institution. State what role the funders took in the study. If the funders had no role in your study, please state: “The funders had no role in study design, data collection and analysis, decision to publish, or preparation of the manuscript.” If any authors received a salary from any of your funders, please state which authors and which funders. If you did not receive any funding for this study, please state: “The authors received no specific funding for this work.”

Response: The authors received no specific funding for this work. The statement has been included within the cover letter.

• Please ensure that you refer to Figure 2 in your text as, if accepted, production will need this reference to link the reader to the figure.

Response: Fig 1 and Fig 2 are now cited under results “Correlation of CSF FII, FIIa and TAT with clinical outcome”.

• Please include captions for your Supporting Information files at the end of your manuscript, and update any in-text citations to match accordingly. Please see our Supporting Information guidelines for more information: http://journals.plos.org/plosone/s/supporting-information.

3. Additional Editor Comments (if provided):

• It would be better to indicate the exact p value rather than showing the range such as p< 0.01, p<0.05, p<0.09 in Table 3 and in the subsequent text.

Response: All p values have been changed to exact values within the manuscript, tables and figures.

Reviewer 1:

Reviewer #1: The authors report high CSF thrombin concentration/activity is associated with an unfavorable outcome with intracerebral hemorrhage, and report thrombin concentration/activity in the CSF of patients with ICH on clinical outcomes in patients presenting with space-occupying lobar supratentorial hemorrhage requiring extra-ventricular drainage (EVD). They measured CSF levels of thrombin, prothrombin, thrombin-antithrombin complexes (TATs), and superoxide dismutase (SOD) in CSF. Initial clot size and intraventricular blood volume were calculated by computerized tomography (CT) on admission, and demographic data, clinical status at admission and neurological outcome were assessed using the modified Rankin Scale (mRS) at 6-weeks and 6-months in 22 consecutive patients (9 females, 11 males) with supratentorial hemorrhage. CSF concentrations of prothrombin (p < 0.005), thrombin (p = 0.005) and TAT (p = 0.046) were statistically different in patients with ICH compared to non-hemorrhagic CSF samples. CSF concentrations of thrombin 24h after ICH correlated with the mRS index after 6 weeks (r2 = 0.73; < 0.005) and 6 months (r2 = 0.63; < 0.005) after discharge from hospital. Thrombin activity, measured via TAT as surrogate parameter of coagulation, likewise correlated with the mRS at 6 weeks (r2 = 0.54; < 0.01) and 6 months (r2 = 0.66; < 0.04). High thrombin concentrations coincide with higher SOD levels 24h after ICH (p = 0.01). They conclude that increased thrombin levels and thrombin activity are associated with a poorer neurological outcome after ICH, and have the potential to serve as predictive markers for individual treatment decision making and future novel therapeutic targets.

Overall, the data and presentation are interesting. The authors are to be congratulated for their interesting study.

Comments:

1. From this reviewer’s perspective, it follows that the larger the clot burden and size of the intracranial hemorrhage, the more probability is that levels of thrombin and other thrombin generation thrombin activity biomarkers will be higher. However, as you thoughtfully noted, intraventricular volumes on CT scans did not correlate with your various biomarkers of thrombin activity in particular thrombin levels. I still think it would be helpful to include a scattergram correlating volume with thrombin levels. Also, has this finding been previously reported?

Response: It is enticing to presume, that larger clot volumes might result in higher CSF thrombin levels. However, given the short half-life of thrombin of about 56.4 ± 4.7 seconds (Ruehl et al. 2012 Thromb haemost.) washed in thrombin might be long gone at the point of analysis. The connection of thrombin and clot volume is now depicted in supplement figure 1. In 2011 Wu et al (European Journal of Neurology) reported a positive correlation of TAT measured in intracerebral hematoma (obtained during surgical removal) and a detrimental outcome. To our knowledge, no correlation between CSF thrombin or TAT levels and clot volume has been reported yet.

2. You state CSF fibrinogen and thoughtfully have control comparisons. Is there any correlation between plasma and CSF fibrinogen levels?

Response: Similar to the absence of a correlation between thrombin and clot volume, we found also no correlation between thrombin levels in CSF and plasma.

3. I’m surprised how small your standard deviation values are for your biomarkers. These are standard deviation and not the standard error of the means, correct?

Response: Figure 2 shows the correlation of thrombin with good and unfavorable outcome at 6 weeks and 6 months after ICH. The box itself represents the interquartile range, while the whiskers indicate the minimum and maximum values. 

4. You use the term ictus, I think to describe stroke or CNS event. I think another descriptive term would be better.

Response: The term ictus has been changed to ICH or onset of ICH respectively.

Additional comments:

Results

5. You state, “Serum levels of FII and FIIa did not correlate with their corresponding CSF levels”. This should be plasma, not serum. You also state this later on as serum levels. Again, coagulation factor levels are obtained from plasma.

Response: The word serum has been replaced by plasma throughout the manuscript.

6. You state , “Further, the TAT-complex as surrogate parameter of coagulation and fibrinolysis correlated with the mRS disability index at 6 weeks (2 = 0.65; p < 0.01) and 6 months (r = 0.65; p < 0.01)”. How does thrombin – antithrombin complexes provide any surrogate parameter of fibrinolysis? Please specifically explain if not remove the term fibrinolysis.

Response: It is true that TAT does actually not allow any conclusion about fibrinolysis. The word fibrinolysis has therefore been removed.

Discussion

7. Your first paragraph restates information that really should be part of your introduction. My suggestion is in your first paragraph state the novelty and important information from your interesting study, and include your second paragraph is your first paragraph.

Response: This is an insightful remark which we gladly adhered to. We have now incorporated the recapitulating information into our introduction with the former second paragraph now being the first of the discussion. Additionally, the introduction was revised and shortened adhering to remarks made by reviewer 2.

8. In your first paragraph, you state, “While small amounts of activated coagulation FII”- be consistent as you described this activated coagulation factor to is thrombin. Is this what you’re referring to? Please be consistent.

Response: The sentence has been deleted due to the restructuring of our introduction. We rechecked the remainder of our manuscript for consistency when referring to FIIa.

9. In your second paragraph, you state, Thrombin is inhibited by antithrombin III, resulting in an inactive proteinase/inhibitor complex (TAT). Interestingly, exceeding concentrations of antithrombin III are also detrimental to neuronal health and might add to adverse thrombin effects.[28]” Antithrombin is primarily a circulating plasma anticoagulant. How does antithrombin get into the CSF? What are the normal levels of antithrombin in the CSF? My understanding is that antithrombin in antithrombin – thrombin complexes are due to active bleeds and breakdown of the blood-brain barrier. Please add this information to your discussion.

Response: This is an insightful remark and important line of thought. Evidence suggests that ATIII plasma levels are 100x higher than in CSF. It is hypothesized that ATIII in CSF is largely derived from passage across the blood-brain barrier. (Zetterberg et al. “CSF Antithrombin III and Disruption of the Blood-Brain Barrier” Journal of Clinical Oncology 2009) This observation falls in line with the mentioned line of thought and has been added to our discussion. 

10. You state, “In our cohort, higher concentrations of thrombin and TAT complex within CSF correlated with a higher mRS at 6 weeks and 6 months after ICH, similar to those of unbound thrombin. These results fall in line with data published from patients with SAH where thrombin activity correlated with the degree of SAH (according to Fisher’s CT classification), the persistence of a subarachnoid clot, and the development of vasospasm.[29]” as previously stated, at least providing a box plot correlation of the intracranial hemorrhage with thrombin generation in your study would be useful.

Response: In order to clarify the interrelation of CSF thrombin and TAT with clot volume, a supplementary figure has been added depicting both together with either intraventricular clot volume or total clot volume.

Reviewer #2: 

1. The introduction could be shortened by 50%.

2. Probably should report variables as median with interquartile range (rather than standard deviation). It's a bit more informative.

Response: The SD in Table 2 has been changed to Median + IQR.

3. How were controls selected? Were they matched for any demographics or comorbidities with patients? If not, how might this limit your results? How was CSF obtained from controls? How might different sampling methods affect your results?

Response: Patients requiring lumbar drainage for normal pressure hydrocephalus evaluation and patients with meningiomas or schwannomas, who need intraoperative opening of CSF spaces were used as controls. This aspect was added to our Methods section. The intention was to compare CSF from patients with ICH to those without. We do not expect that thrombin or ATIII concentration would differ throughout the CSF. Nevertheless, to our knowledge, no comparison of these factors according to site of CSF acquisition has been reported so far. The controls were similar in age and sex distribution. As comorbidities were widely spread among patients with ICH (Supp. Table 1), arterial hypertension was the sole common denominator. 

4. I could not find mention of whether any patients had intraventricular hematoma, and if so, how much?

Response: All patients had at least minimal aspects of intraventricular hemorrhage. The mean (SD) intraventricular blood volume is shown in table 1. An explanatory graph of intraventricular clot and thrombin/TAT correlation has been added as supp. Fig.1

5. Need consistency in the number of decimal places used to report out p-values in tables.

Response: Number and p-value formatting has been revised throughout the manuscript.

6. Please report mean/median time to CSF sample collection from ICH ictus, if you have it, or list this as a limitation.

Response: This is an import observation. As our data concerning the events prior to hospital admission are limited, this circumstance has been added as limitation of our study.

7. Given the very small sample size, the conclusion of both the abstract and the manuscript are over-stated. It is possible that CSF thrombin may be associated with outcome, but it's also quite possible that a larger, better powered, more random sample would prove your current findings wrong. Or that a larger study would find that only certain patients (perhaps with certain hematomas) demonstrate the associations that you found. Alot left to learn and your data are very preliminary.

Response: We agree that conclusion from this small and heterogenous cohort have to be drawn with caution and therefor somewhat dampened the conclusions in our manuscript.

8. What do you think of your findings in the context of the translational science work using preclinical SAH models - Wan, S., Wei, J., Hua, Y. et al. Cerebrospinal Fluid from Aneurysmal Subarachnoid Hemorrhage Patients Leads to Hydrocephalus in Nude Mice. Neurocrit Care (2020). https://doi.org/10.1007/s12028-020-01031-0?

Response: This is an interesting study suggesting that various a-cellular components of CSF might influence secondary injury formation via macrophage activation and consecutive cell damage after SAH. These results might be transferable to intraventricular hemorrhages altering CSF composition and thus effecting secondary injury.

Overall, nicely done.

---

## [Decision Letter · Decision Letter 1]

19 Oct 2020

High CSF thrombin concentration and activity is associated with an unfavorable outcome in patients with intracerebral hemorrhage

PONE-D-20-21960R1

Dear Dr. Krenzlin,

We’re pleased to inform you that your manuscript has been judged scientifically suitable for publication and will be formally accepted for publication once it meets all outstanding technical requirements.

Kind regards,

Tamil Selvan Anthonymuthu, Ph. D

Academic Editor

PLOS ONE

Additional Editor Comments (optional):

Reviewers' comments:

Reviewer's Responses to Questions

**Comments to the Author**

1. If the authors have adequately addressed your comments raised in a previous round of review and you feel that this manuscript is now acceptable for publication, you may indicate that here to bypass the “Comments to the Author” section, enter your conflict of interest statement in the “Confidential to Editor” section, and submit your "Accept" recommendation.

Reviewer #1: All comments have been addressed

Reviewer #2: (No Response)

2. Is the manuscript technically sound, and do the data support the conclusions?

Reviewer #1: Yes

Reviewer #2: (No Response)

3. Has the statistical analysis been performed appropriately and rigorously? 

Reviewer #1: I Don't Know

Reviewer #2: (No Response)

4. Have the authors made all data underlying the findings in their manuscript fully available?

Reviewer #1: Yes

Reviewer #2: (No Response)

5. Is the manuscript presented in an intelligible fashion and written in standard English?

Reviewer #1: Yes

Reviewer #2: (No Response)

6. Review Comments to the Author

Reviewer #1: The authors have done an excellent job of revising their manuscript, answering the comments, and providing important responses. I have no further comments.

Reviewer #2: Well done. All of my concerns have been addressed.

For some reason, PLOS ONE requires a character count for these statements, which I have now met.

7. PLOS authors have the option to publish the peer review history of their article (what does this mean?). If published, this will include your full peer review and any attached files.

Reviewer #1: No

Reviewer #2: No

---

## [Editor Report · Acceptance letter]

3 Nov 2020

PONE-D-20-21960R1 

High CSF thrombin concentration and activity is associated with an unfavorable outcome in patients with intracerebral hemorrhage 

Dear Dr. Krenzlin:

I'm pleased to inform you that your manuscript has been deemed suitable for publication in PLOS ONE. Congratulations! Your manuscript is now with our production department. 

Kind regards, 

on behalf of

Dr. Tamil Selvan Anthonymuthu 

Academic Editor

PLOS ONE